# Blood glucose fluctuation and in-hospital mortality among patients with acute myocardial infarction: eICU collaborative research database

**Junhua Chen, Weifang Huang, Nan Liang**  *

Department of Cardiology, Xinjiang Armed Police Corps Hospital, Urumqi City, Xinjiang Province, P. R. China

* liangnandct@outlook.com

## Abstract

### Background

To assess the relationship between glycemic variability, glucose fluctuation trajectory and the risk of in-hospital mortality in patients with acute myocardial infarction (AMI).

### Methods

This retrospective cohort study included AMI patients from eICU Collaborative Research Database. In-hospital mortality of AMI patients was primary endpoint. Blood glucose levels at admission, glycemic variability, and glucose fluctuation trajectory were three main study variables. Blood glucose levels at admission were stratified into: normal, intermediate, and high. Glycemic variability was evaluated using the coefficient of variation (CV), divided into four groups based on quartiles: quartile 1: CV≤10; quartile 2: 10<CV≤20; quartile 3: 20<CV≤30; quartile 4: CV>30. Univariate and multivariate Cox regression models to assess the relationship between blood glucose levels at admission, glycemic variability, glucose fluctuation trajectory, and in-hospital mortality in patients with AMI.

### Results

2590 participants were eventually included in this study. There was a positive relationship between high blood glucose level at admission and in-hospital mortality [hazard ratio (HR) = 1.42, 95%confidence interval (CI): 1.06–1.89]. The fourth quartile (CV>30) of CV was associated with increased in-hospital mortality (HR = 2.06, 95% CI: 1.25–3.40). The findings indicated that only AMI individuals in the fourth quartile of glycemic variability, exhibited an elevated in-hospital mortality among those with normal blood glucose levels at admission (HR = 2.33, 95% CI: 1.11–4.87). Additionally, elevated blood glucose level was a risk factor for in-hospital mortality in AMI patients.

**Funding:** The author(s) received no specific funding for this work.

**Competing interests:** The authors have declared that no competing interests exist.

## Conclusion

Glycemic variability was correlated with in-hospital mortality, particularly among AMI patients who had normal blood glucose levels at admission. Our study findings also suggest early intervention should be implemented to normalize high blood glucose levels at admission of AMI.

## Introduction

Acute myocardial infarction (AMI) is a fatal disease caused by acute and persistent ischemia and hypoxia of coronary arteries, resulting in high morbidity and mortality rates [1]. Exploring simple biomarkers is crucial for assessing the risk of in-hospital mortality in patients with AMI.

A rising body of studies has suggested that elevated glucose levels at admission are frequently observed in patients with AMI, which have also been linked to a poorer prognosis during hospitalization and post-discharge [1–3]. In the study of Ferreira et al., they concluded that admission hyperglycemia was associated with higher all-cause mortality in patients with AMI, regardless of previous diabetic status [4]. Admission hyperglycemia may be linked to an unfavorable prognosis in patients with AMI due to microvascular injury precipitating no-reflow phenomenon [5]. However, intensive the insulin therapy is associated with an increased risk of unfavorable outcomes, potentially attributed to fluctuations in glucose levels and occurrences of hypoglycemia [6]. Furthermore, several research have indicated that focusing on blood glucose fluctuations may offer superior clinical management compared to solely monitoring [7,8]. Glycemic variability, oscillations in blood glucose levels, refers to the measurement of fluctuations in glucose or other related parameters of glucose homeostasis over a specific time period [9]. An elevated glycemic variability can contribute to the activation of oxidative stress, impairment of endothelial function, and glycosylation of proteins [10–12]. The findings of previous studies have demonstrated a positive correlation between glycemic variability and mortality in critically ill patients, including those with subarachnoid hemorrhage and sepsis [13,14]. Notably, no previously published study has examined the association between blood glucose fluctuations or variability and in-hospital mortality among patients diagnosed with AMI.

Herein, the objective of this study was to analyze the correlation of glycemic variability and glucose fluctuation trajectory with the risk of in-hospital mortality in patients with AMI, exploring a reasonable range for short-term blood glucose control in patients with AMI and providing certain basis for treatment decision-making and prognosis improvement.

## Methods

### Data source

Data for our analysis was sourced from the emergency intensive care unit (eICU) collaborative research database. The eICU collaborative research database is a large public database developed by the eICU Research Institute in collaboration with the Laboratory for Computational Physiology at Massachusetts Institute of Technology, which encompassing de-identified data from 200,859 intensive care unit (ICU) stays across 208 hospitals in the United States between 2014–2015 [15]. The utilization of this database has been granted approval by the Institutional Review Board at the Massachusetts Institute of Technology. All patient identity information is anonymized, thereby obviating the need for patient informed consent.

## Study population

In this retrospective cohort study, we included patients from the eICU collaborative research database who diagnosed as AMI according to the International Classification of Diseases (ICD) code (ICD code: 140) at ICU admission and had at least two glucose measurement within 48 hours after ICU admission. The following exclusion criteria were used: (1) age <18 years old; (2) the length of ICU stay <48 hours; (3) missing survival data. Fig 1 shows the complete inclusion and exclusions processes of study population.

## Endpoint

In-hospital mortality among patients with AMI was primary endpoint in this study. The patient cohort was categorized into two groups by survival status: survivors and non-survivors. The median follow-up time was 6.32 (3.99, 9.83) days.

## Main study variable

Baseline (at ICU admission) blood glucose level, glycemic variability within two days after ICU admission, and glucose fluctuation trajectory within two days after ICU admission were three main study variables for this study. Blood glucose levels at admission were stratified into three levels: normal (<140 mg/dL), intermediate (140–200 mg/dL), and high (≥200 mg/dL) [16]. Glycemic variability was evaluated using the coefficient of variation (CV), which is calculated as the ratio of the standard deviation of blood glucose measurements to the mean value. In this study, we divided CV into four groups based on quartiles: quartile 1: CV≤10; quartile 2: 10<CV≤20; quartile 3: 20<CV≤30; quartile 4: CV>30. Glucose fluctuation trajectory was defined as the fluctuation direction between blood glucose level at admission and the last measured blood glucose level within two days of admission to the ICU. "Normal to Normal" means that both baseline and last measured blood glucose levels were "Normal"; "Intermediate

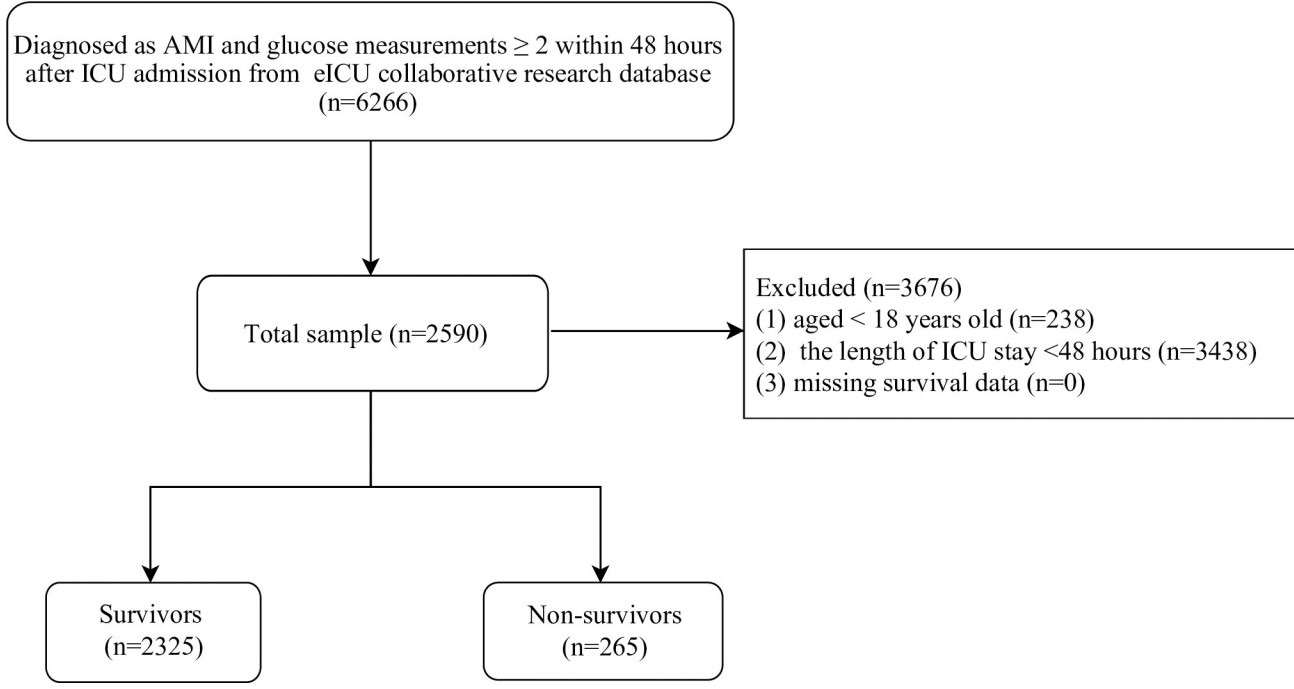

**Fig 1. Flow chart of patients' enrollment process.**

to Intermediate" indicates that both baseline and last measured blood glucose levels were "Intermediate"; "High to High" means that both baseline and last measured blood glucose levels are "High"; "UP" indicates that the blood glucose level last measured was elevated compared to blood glucose level at admission; "High or Intermediate to Normal" indicates that the blood glucose level at admission was "High" or "Intermediate", and the last measured blood glucose level was "Normal"; "High to Intermediate" means that the blood glucose at admission is "High" and the last measured blood glucose level is "Intermediate".

## Data collection

The following variables on admission of patients were extracted from the eICU collaborative research database: age (years), gender, ethnicity, ICU type, comorbidity (including congestive heart failure, atrial fibrillation, hypertension, diabetes, cardiogenic shock, sepsis), height (cm), weight (kg), body mass index (BMI, kg/m$^2$), vital signs [including heart rate (beats/minute), systolic blood pressure (SBP, mmHg), diastolic blood pressure (DBP, mmHg), respiratory rate (times/min), temperature (˚C)], laboratory parameters [including white blood cell (WBC, K/mcL), platelet (K/mcL), hemoglobin (g/dL), red blood cell distribution width (RDW, %), creatinine (mg/dL), estimated glomerular filtration rate (eGFR) by chronic kidney disease epidemiology collaboration equation (eGFR$_{CKD-EPI}$, mL/min/1.73m$^2$), bicarbonate (mmol/L), potassium (mmol/L), sodium (mmol/L), chloride (mmol/L)], treatment [mechanical ventilation (MV), vasopressor use, percutaneous coronary intervention (PCI), coronary artery bypass graft (CABG), thrombolysis, insulin, angiotensin converting enzyme (ACE) inhibitor, angiotensin II receptor blocker (ARB), anticoagulant administration, antiplatelet agent, beta blocker, calcium channel blocker].

## Statistical analysis

Descriptive statistics: Mean ± standard deviation (Mean ± SD) and median and quartiles [M (Q1, Q3)] were utilized to describe the distribution of measurement data, whether normally or non-normally distributed. The differences between groups were compared using the t-test and Wilcoxon rank sum test. The categorical data were presented as the number of cases and the constituent ratio [N (%)], and group comparisons were performed using the Chi-square tests or Fisher precision tests. The variables with missing values exceeding 20% were excluded, while the variables with missing values below 20% were imputed using the multiple imputation method. Sensitivity analysis before and after interpolation was performed (S1 Table).

Univariate and multivariate Cox stepwise regression analyses was used to screen covariates (S2 Table). Then, we used univariate and multivariate Cox regression models to assess the relationship between blood glucose levels at admission, glycemic variability, glucose fluctuation trajectory, and the risk of hospital mortality in patients with AMI. Model 1 was univariate Cox regression model (unadjusted variables); Model 2 adjusted age, hypertension, cardiogenic shock, respiratory rate, RDW, creatinine, sodium, mechanical ventilation, vasopressor use, and CABG. Hazard ratio (HR) with 95% confidence interval (CI) was calculated. *P* value<0.05 was considered significant for all analyses. Statistical analysis was performed using PostgreSQL 14.7, SAS 9.4 and R 4.2.3.

## Results

### Characteristics of the study participants

Overall, 2590 participants in the eICU collaborative research database were eventually included in this study (Fig 1). The mean (± SD) age of included participants was 67.22 (±

12.40) years, and 40.31% were female. (Table 1). Most of the participants were Caucasian (77.92%). The blood glucose levels at admission of 46.41% of the participants were within the normal range, while 26.95% had intermediate levels and 26.64% had high levels. In addition, 265 patients died and 2325 survived, respectively. The comparison of baseline characteristics, vital signs, laboratory parameters, and treatment between survivors and non-survivors was also summarized in Table 1. The non-survivors, in comparison to the survivors, exhibited advanced age and a higher proportion of congestive heart failure, atrial fibrillation, cardiogenic shock, sepsis. Additionally, they had higher levels of heart rate, respiratory rate, WBC, RDW, and creatinine.

## Blood glucose at admission and in-hospital mortality

The association of blood glucose at admission and in-hospital mortality in patients with AMI was listed in Table 2. In univariate analysis, taking normal levels as the reference group, high blood glucose level at admission was associated with increased in-hospital mortality (Model 1: HR = 1.76, 95%CI: 1.33–2.32, $P<0.001$). After adjusting age, hypertension, cardiogenic shock, respiratory rate, RDW, creatinine, sodium, mechanical ventilation, vasopressor use, and CABG, high blood glucose level at admission was still shown to be positively correlated with in-hospital mortality (Model 2: HR = 1.42, 95%CI: 1.06–1.89, $P = 0.019$).

## Glycemic variability and in-hospital mortality

Simultaneously, Table 2 also revealed the association of glycemic variability and in-hospital mortality in patients with AMI. In the Model 1, compared with the first quartile of CV ($\leq$10), the risk of in-hospital mortality in the second quartile (10<CV$\leq$20; HR = 1.89, 95% CI: 1.15–3.11, $P = 0.012$), the third quartile (20<CV$\leq$30; HR = 1.89, 95% CI: 1.12–3.16, $P = 0.016$), and the fourth quartile (CV>30; HR = 2.99, 95% CI: 1.83–4.88, $P<0.001$) increased significantly. In the fully-adjusted model, we observed that only the fourth quartile (CV>30) of CV was associated with increased in-hospital mortality (HR = 2.06, 95% CI: 1.25–3.40, $P = 0.004$).

To further investigate the correlation between glycemic variability and the risk of in-hospital mortality in a population with varying blood glucose levels at admission, we conducted a subgroup analysis based on individuals with different blood glucose levels at admission (Table 3). The glycemic variability at different blood glucose levels at admission is illustrated in Fig 2, demonstrating an increase in short-term glycemic variability as the blood glucose level at admission rises. As shown in Table 3, the findings only indicated that only AMI individuals in the fourth quartile of glycemic variability, with the first quartile serving as the reference group, exhibited an elevated risk of in-hospital mortality among those with normal blood glucose levels at admission (HR = 2.33, 95% CI: 1.11–4.87, $P = 0.025$).

## Glucose fluctuation trajectory and in-hospital mortality

Table 4 shows the association of glucose fluctuation trajectory and in-hospital mortality. The HR value (95% CI) of UP group, with reference to the NN group, was 1.73 (1.16–2.59), indicating that elevated blood glucose level was a significant risk factor for in-hospital mortality in patients with AMI ($P = 0.007$). Additionally, the HR values (95% CI) in the "High to High" and"High to Intermediate" groups were 1.76 (1.04–2.98) and 2.01 (1.32–3.07), respectively, indicating that high blood glucose levels at admission increased the risk of in-hospital mortality. Despite implementing effective measures to reduce blood glucose to intermediate levels, there remains a high in-hospital mortality (HR = 2.01, 95%CI: 1.32–3.07, $P = 0.001$). These findings suggest that when blood glucose levels at admission are high, early action may be taken to reduce them to normal levels.

**Table 1. Characteristics of the study participants.**

| Variables | Total (n = 2590) | Survivors (n = 2325) | Non-survivors (n = 265) | Statistics | *P* |
|---|---|---|---|---|---|
| **General characteristics** | | | | | |
| Age, year, Mean ± SD | 67.22 ± 12.40 | 66.88 ± 12.43 | 70.16 ± 11.73 | t = -4.09 | <0.001 |
| Gender, n (%) | | | | $\chi^2$ = 1.169 | 0.280 |
| Female | 1044 (40.31) | 929 (39.96) | 115 (43.40) | | |
| Male | 1546 (59.69) | 1396 (60.04) | 150 (56.60) | | |
| Ethnicity, n (%) | | | | $\chi^2$ = 7.083 | 0.132 |
| African American | 242 (9.34) | 223 (9.59) | 19 (7.17) | | |
| Asian | 52 (2.01) | 47 (2.02) | 5 (1.89) | | |
| Caucasian | 2018 (77.92) | 1802 (77.51) | 216 (81.51) | | |
| Hispanic | 79 (3.05) | 67 (2.88) | 12 (4.53) | | |
| Other/Unknown | 199 (7.68) | 186 (8.00) | 13 (4.91) | | |
| ICU type, n (%) | | | | $\chi^2$ = 9.223 | 0.237 |
| CCU-CTICU | 482 (18.61) | 440 (18.92) | 42 (15.85) | | |
| CSICU | 148 (5.71) | 125 (5.38) | 23 (8.68) | | |
| CTICU | 106 (4.09) | 97 (4.17) | 9 (3.40) | | |
| CCU | 406 (15.68) | 358 (15.40) | 48 (18.11) | | |
| MICU | 118 (4.56) | 108 (4.65) | 10 (3.77) | | |
| MICU-SICU | 1204 (46.49) | 1083 (46.58) | 121 (45.66) | | |
| NICU | 52 (2.01) | 45 (1.94) | 7 (2.64) | | |
| SICU | 74 (2.86) | 69 (2.97) | 5 (1.89) | | |
| Congestive heart failure, n (%) | | | | $\chi^2$ = 13.729 | <0.001 |
| No | 2037 (78.65) | 1852 (79.66) | 185 (69.81) | | |
| Yes | 553 (21.35) | 473 (20.34) | 80 (30.19) | | |
| Atrial fibrillation, n (%) | | | | $\chi^2$ = 6.435 | 0.011 |
| No | 2250 (86.87) | 2033 (87.44) | 217 (81.89) | | |
| Yes | 340 (13.13) | 292 (12.56) | 48 (18.11) | | |
| Hypertension, n (%) | | | | $\chi^2$ = 3.210 | 0.073 |
| No | 2114 (81.62) | 1887 (81.16) | 227 (85.66) | | |
| Yes | 476 (18.38) | 438 (18.84) | 38 (14.34) | | |
| Diabetes, n (%) | | | | $\chi^2$ = 0.627 | 0.429 |
| No | 2052 (79.23) | 1847 (79.44) | 205 (77.36) | | |
| Yes | 538 (20.77) | 478 (20.56) | 60 (22.64) | | |
| Cardiogenic shock, n (%) | | | | $\chi^2$ = 84.998 | <0.001 |
| No | 2303 (88.92) | 2112 (90.84) | 191 (72.08) | | |
| Yes | 287 (11.08) | 213 (9.16) | 74 (27.92) | | |
| Sepsis, n (%) | | | | $\chi^2$ = 38.765 | <0.001 |
| No | 2158 (83.32) | 1973 (84.86) | 185 (69.81) | | |
| Yes | 432 (16.68) | 352 (15.14) | 80 (30.19) | | |
| **Physical examination** | | | | | |
| Height, cm, Mean ± SD | 169.65 ± 10.42 | 169.73 ± 10.45 | 168.96 ± 10.19 | t = 1.13 | 0.257 |
| Weight, kg, Mean ± SD | 83.88 ± 21.40 | 84.01 ± 21.28 | 82.70 ± 22.37 | t = 0.94 | 0.346 |
| BMI, kg/m², Mean ± SD | 29.08 ± 6.82 | 29.09 ± 6.75 | 28.94 ± 7.41 | t = 0.33 | 0.744 |
| **Vital signs** | | | | | |
| Heart rate, beats/minute, Mean ± SD | 88.69 ± 22.45 | 88.12 ± 22.57 | 93.67 ± 20.80 | t = -3.82 | <0.001 |
| SBP, mmHg, Mean ± SD | 126.44 ± 29.49 | 127.31 ± 29.20 | 118.80 ± 30.92 | t = 4.47 | <0.001 |
| DBP, mmHg, Mean ± SD | 72.30 ± 19.22 | 72.67 ± 19.11 | 69.08 ± 19.84 | t = 2.89 | 0.004 |
| Respiratory rate, times/min, M (Q₁, Q₃) | 20.00 (16.00, 24.00) | 19.00 (16.00, 23.00) | 20.00 (18.00, 25.00) | Z = 4.177 | <0.001 |

*(Continued)*

**Table 1.** (*Continued*)

| Variables | Total (n = 2590) | Survivors (n = 2325) | Non-survivors (n = 265) | Statistics | *P* |
|---|---|---|---|---|---|
| Temperature, ˚C, Mean ± SD | 36.98 ± 4.71 | 36.93 ± 4.23 | 37.34 ± 7.77 | t = -0.84 | 0.400 |
| **Laboratory parameters** | | | | | |
| WBC, K/mcL, M (Q$_1$, Q$_3$) | 11.73 (8.90, 15.80) | 11.53 (8.80, 15.32) | 13.80 (10.50, 18.80) | Z = 5.444 | <0.001 |
| Platelets, K/mcL, M (Q$_1$, Q$_3$) | 217.00 (168.00, 274.00) | 218.00 (169.00, 273.00) | 211.00 (154.00, 274.00) | Z = -1.238 | 0.216 |
| Hemoglobin, g/dL, Mean ± SD | 12.32 ± 2.57 | 12.37 ± 2.57 | 11.87 ± 2.47 | t = 3.00 | 0.003 |
| RDW, %, Mean ± SD | 14.70 ± 1.97 | 14.64 ± 1.92 | 15.29 ± 2.30 | t = -4.48 | <0.001 |
| Creatinine, mg/dL, M (Q$_1$, Q$_3$) | 1.16 (0.88, 1.70) | 1.12 (0.86, 1.66) | 1.50 (1.10, 2.34) | Z = 7.345 | <0.001 |
| eGFR$_{CKD-EPI}$, mL/min/1.73m$^2$, M (Q$_1$, Q$_3$) | 65.04 (41.41, 86.77) | 67.10 (43.05, 88.00) | 48.24 (28.35, 66.59) | Z = -7.949 | <0.001 |
| Bicarbonate, mmol/L, Mean ± SD | 23.38 ± 4.93 | 23.51 ± 4.84 | 22.27 ± 5.54 | t = 3.50 | <0.001 |
| Sodium, mmol/L, Mean ± SD | 137.42 ± 5.01 | 137.35 ± 4.83 | 138.07 ± 6.34 | t = -1.78 | 0.077 |
| Potassium, mmol/L, Mean ± SD | 4.19 ± 0.78 | 4.17 ± 0.76 | 4.35 ± 0.89 | t = -3.09 | 0.002 |
| Chloride, mmol/L, Mean ± SD | 102.77 ± 6.34 | 102.73 ± 6.14 | 103.16 ± 7.88 | t = -0.87 | 0.387 |
| Blood glucose at admission, n (%) | | | | $\chi^2$ = 28.776 | <0.001 |
| Normal | 1202 (46.41) | 1110 (47.74) | 92 (34.72) | | |
| Intermediate | 698 (26.95) | 631 (27.14) | 67 (25.28) | | |
| High | 690 (26.64) | 584 (25.12) | 106 (40.00) | | |
| CV, n (%) | | | | $\chi^2$ = 47.708 | <0.001 |
| Quartile 1 | 455 (17.57) | 436 (18.75) | 19 (7.17) | | |
| Quartile 2 | 912 (35.21) | 828 (35.61) | 84 (31.70) | | |
| Quartile 3 | 615 (23.75) | 555 (23.87) | 60 (22.64) | | |
| Quartile 4 | 608 (23.47) | 506 (21.76) | 102 (38.49) | | |
| **Treatment** | | | | | |
| Mechanical ventilation, n (%) | | | | $\chi^2$ = 139.611 | <0.001 |
| No | 1632 (63.01) | 1553 (66.80) | 79 (29.81) | | |
| Yes | 958 (36.99) | 772 (33.20) | 186 (70.19) | | |
| Vasopressor use, n (%) | | | | $\chi^2$ = 125.188 | <0.001 |
| No | 1825 (70.46) | 1717 (73.85) | 108 (40.75) | | |
| Yes | 765 (29.54) | 608 (26.15) | 157 (59.25) | | |
| PCI, n (%) | | | | $\chi^2$ = 2.216 | 0.137 |
| No | 2091 (80.73) | 1868 (80.34) | 223 (84.15) | | |
| Yes | 499 (19.27) | 457 (19.66) | 42 (15.85) | | |
| CABG, n (%) | | | | $\chi^2$ = 7.855 | 0.005 |
| No | 2389 (92.24) | 2133 (91.74) | 256 (96.60) | | |
| Yes | 201 (7.76) | 192 (8.26) | 9 (3.40) | | |
| Thrombolysis, n (%) | | | | - | 0.117 |
| No | 2560 (98.84) | 2301 (98.97) | 259 (97.74) | | |
| Yes | 30 (1.16) | 24 (1.03) | 6 (2.26) | | |
| Insulin, n (%) | | | | $\chi^2$ = 3.254 | 0.071 |
| No | 1982 (76.53) | 1791 (77.03) | 191 (72.08) | | |
| Yes | 608 (23.47) | 534 (22.97) | 74 (27.92) | | |
| ACE inhibitor, n (%) | | | | $\chi^2$ = 4.738 | 0.029 |
| No | 2421 (93.47) | 2165 (93.12) | 256 (96.60) | | |
| Yes | 169 (6.53) | 160 (6.88) | 9 (3.40) | | |
| ARB, n (%) | | | | $\chi^2$ = 1.950 | 0.163 |
| No | 2573 (99.34) | 2308 (99.27) | 265 (100.00) | | |
| Yes | 17 (0.66) | 17 (0.73) | 0 (0.00) | | |
| Anticoagulant administration, n (%) | | | | $\chi^2$ = 0.630 | 0.427 |

(*Continued*)

**Table 1.** (Continued)

| Variables | Total (n = 2590) | Survivors (n = 2325) | Non-survivors (n = 265) | Statistics | *P* |
|---|---|---|---|---|---|
| No | 1942 (74.98) | 1738 (74.75) | 204 (76.98) | | |
| Yes | 648 (25.02) | 587 (25.25) | 61 (23.02) | | |
| Antiplatelet agent, n (%) | | | | $\chi^2 = 2.759$ | 0.097 |
| No | 1780 (68.73) | 1586 (68.22) | 194 (73.21) | | |
| Yes | 810 (31.27) | 739 (31.78) | 71 (26.79) | | |
| Beta blocker, n (%) | | | | $\chi^2 = 7.536$ | 0.006 |
| No | 2140 (82.63) | 1905 (81.94) | 235 (88.68) | | |
| Yes | 450 (17.37) | 420 (18.06) | 30 (11.32) | | |
| Calcium channel blocker, n (%) | | | | - | 0.487 |
| No | 2568 (99.15) | 2306 (99.18) | 262 (98.87) | | |
| Yes | 22 (0.85) | 19 (0.82) | 3 (1.13) | | |

ICU = intensive care unit; CCU = coronary heart disease intensive care unit; CTICU = cardiothoracic intensive care unit; CSICU = cardiac surgery intensive care unit; MICU = medical intensive care unit; SICU = surgical intensive care unit; NICU = neurosurgical intensive care unit; BMI = body mass index; SBP = systolic blood pressure; DBP = diastolic blood pressure; WBC = white blood cell count; RDW = red blood cell distribution width; $eGFR_{CKD-EPI}$ = estimated glomerular filtration rate by chronic kidney disease epidemiology collaboration equation; CV = coefficient of variation; PCI = percutaneous coronary intervention; CABG = coronary artery bypass graft; ACE = angiotensin converting enzyme; ARB = angiotensin II receptor blocker.

## Discussion

This retrospective cohort study investigated the relationship between blood glucose levels at admission, glycemic variability, glucose fluctuation trajectory, and the risk of in-hospital mortality in patients with AMI. The findings indicated that highest quartile (CV>30) of CV were found to be positively correlated with in-hospital mortality, particularly in AMI patients with normal blood glucose levels at admission. In addition, the presence of elevated blood glucose level at admission of AMI patients was observed as a significant risk factor for in-hospital mortality. Our result also indicated that prompt management of elevated blood glucose levels upon admission and maintaining it within the normal range may be advantageous for the prognosis of patients with AMI.

**Table 2. Blood glucose at admission, glycemic variability, and in-hospital mortality.**

| Variables | Model 1 | | Model 2 | |
|---|---|---|---|---|
| | HR (95%*CI*) | *P* | HR (95%CI) | *P* |
| Blood glucose at admission | | | | |
| Normal | Ref | | Ref | |
| Intermediate | 1.16 (0.84–1.58) | 0.369 | 1.01 (0.73–1.40) | 0.941 |
| High | 1.76 (1.33–2.32) | <0.001 | 1.42 (1.06–1.89) | 0.019 |
| CV | | | | |
| Quartile 1 | Ref | | Ref | |
| Quartile 2 | 1.89 (1.15–3.11) | 0.012 | 1.59 (0.96–2.62) | 0.070 |
| Quartile 3 | 1.89 (1.12–3.16) | 0.016 | 1.46 (0.87–2.46) | 0.151 |
| Quartile 4 | 2.99 (1.83–4.88) | <0.001 | 2.06 (1.25–3.40) | 0.004 |

CV = coefficient of variation; HR = hazard ratio; CI = confidence interval.
Model 1: Was unadjusted model.
Model 2: Adjusted age, hypertension, cardiogenic shock, respiratory rate, red blood cell distribution width, creatinine, sodium, mechanical ventilation, vasopressor use, and coronary artery bypass graft.

**Table 3. Relationship of glycemic variability and in-hospital mortality based on different blood glucose levels at admission.**

| CV | Normal | | | Intermediate | | | High | | |
|---|---|---|---|---|---|---|---|---|---|
| | Sample size | HR (95%CI) | P | Sample size | HR (95%CI) | P | Sample size | HR (95%CI) | P |
| Quartile 1 | n = 360 | Ref | | n = 76 | Ref | | n = 19 | Ref | |
| Quartile 2 | n = 478 | 1.39 (0.69–2.77) | 0.356 | n = 305 | 1.27 (0.55–2.93) | 0.577 | n = 129 | 2.59 (0.34–19.43) | 0.355 |
| Quartile 3 | n = 216 | 1.60 (0.75–3.44) | 0.226 | n = 207 | 1.11 (0.45–2.77) | 0.816 | n = 192 | 1.75 (0.23–13.12) | 0.584 |
| Quartile 4 | n = 148 | 2.33 (1.11–4.87) | 0.025 | n = 110 | 1.41 (0.54–3.68) | 0.483 | n = 350 | 2.06 (0.28–15.06) | 0.475 |

CV = coefficient of variation; HR = hazard ratio; CI = confidence interval.

Adjusted age, hypertension, cardiogenic shock, respiratory rate, red blood cell distribution width, creatinine, sodium, mechanical ventilation, vasopressor use, and coronary artery bypass graft.

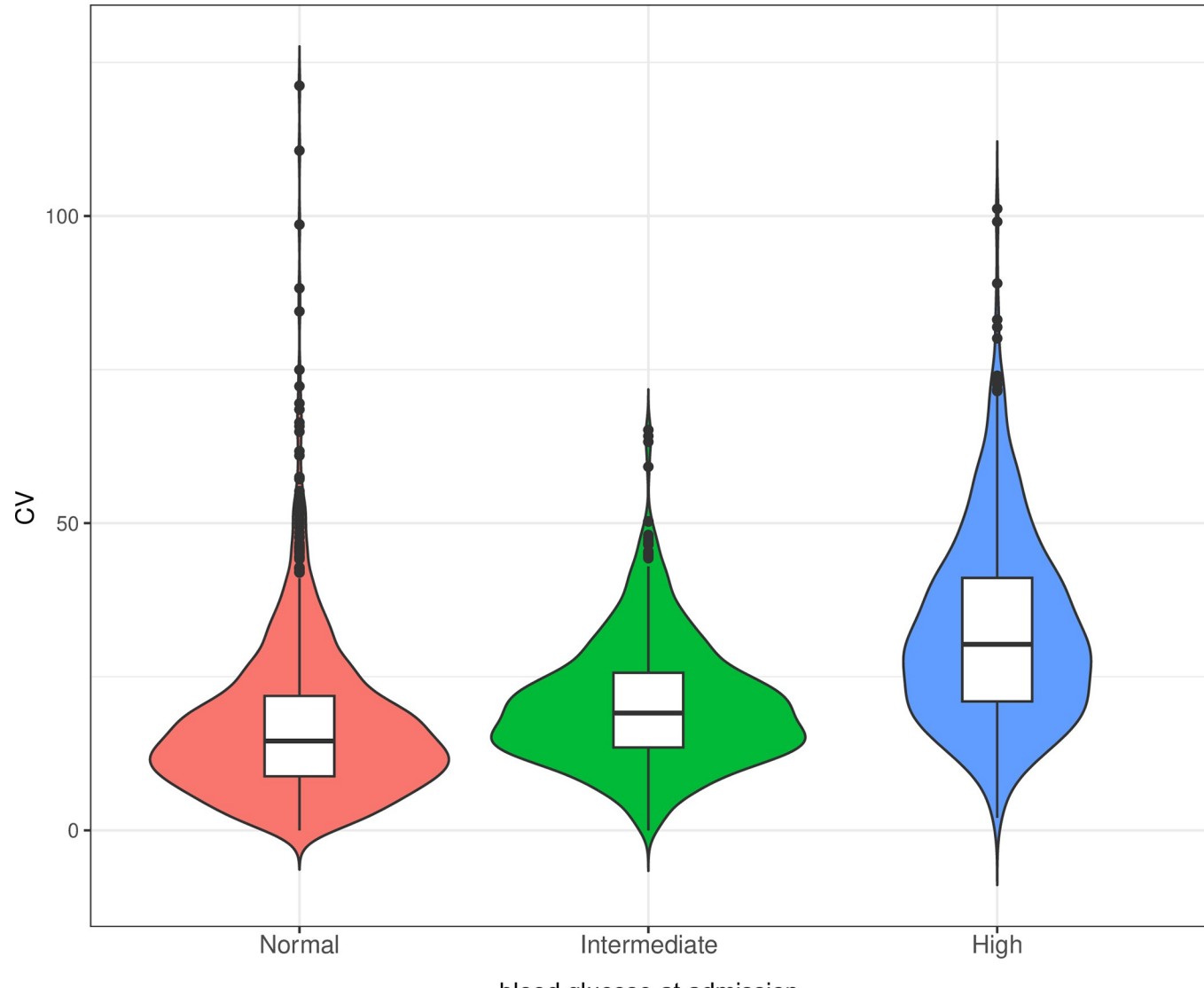

**Fig 2. The glycemic variability at different blood glucose levels at admission.**

**Table 4. Glucose fluctuation trajectory and in-hospital mortality.**

| Glucose fluctuation trajectory | Sample size, n (%) | Model 1 | | Model 2 | |
|---|---|---|---|---|---|
| | | HR (95%CI) | P | HR (95%CI) | P |
| "Normal to Normal" | 932 (35.98) | Ref | | Ref | |
| "Intermediate to Intermediate" | 221 (8.53) | 1.80 (1.10–2.96) | 0.020 | 1.45 (0.88–2.39) | 0.145 |
| "High to High" | 192 (7.41) | 1.87 (1.11–3.14) | 0.019 | 1.76 (1.04–2.98) | 0.035 |
| "UP" | 346 (13.36) | 2.27 (1.53–3.36) | <0.001 | 1.73 (1.16–2.59) | 0.007 |
| "High or Intermediate to Normal" | 629 (24.29) | 1.91 (1.33–2.74) | <0.001 | 1.37 (0.95–1.98) | 0.095 |
| "High to Intermediate" | 270 (10.42) | 2.63 (1.74–3.97) | <0.001 | 2.01 (1.32–3.07) | 0.001 |

HR = hazard ratio; CI = confidence interval.

"Normal to Normal" means that both baseline and last measured blood glucose levels were "Normal".

"Intermediate to Intermediate" indicates that both baseline and last measured blood glucose levels were "Intermediate".

"High to High" means that both baseline and last measured blood glucose levels are "High".

"UP" indicates that the blood glucose level last measured was elevated compared to blood glucose level at admission.

"High or Intermediate to Normal" indicates that the blood glucose level at admission was "High" or "Intermediate", and the last measured blood glucose level was "Normal".

"High to Intermediate" means that the blood glucose at admission is "High" and the last measured blood glucose level is "Intermediate".

Model 1: Was unadjusted model.

Model 2: Adjusted age, hypertension, cardiogenic shock, respiratory rate, red blood cell distribution width, creatinine, sodium, mechanical ventilation, vasopressor use, and coronary artery bypass graft.

Previous clinical studies have demonstrated that patients with AMI were often accompanied by elevated glucose levels upon admission, which has been found to be associated with poorer prognosis [17,18]. Consistent with prior research findings [19,20], this study also corroborated the observation that high blood glucose levels at ICU admission was a risk factor for in-hospital mortality in AMI patients after adjusting all confounding factors. The presence of high blood glucose levels in critically ill patients with AMI has the potential to initiate inflammation and oxidative stress, worsen endothelial dysfunction, and prompt a pre-thrombotic state, which ultimately results in impaired coronary blood flow and an increased size of the infarct [20,21]. There is mounting evidence suggesting that glycemic variability may possess prognostic value for certain cardiovascular events [22,23]. In the development of complications related to cardiovascular disease, fluctuations in glucose levels seem to exert a more detrimental impact than persistent hyperglycemia [24]. This may be due to the fact that fluctuating blood glucose levels induce greater oxidative stress than consistently high blood glucose, specifically by accelerating the production of super-oxides in mitochondrial and vascular inflammation [25]. However, it remains unclear whether glycemic variability holds prognostic significance for in-hospital mortality of AMI patients. Our findings indicate a significant association between glycemic variability and in-hospital mortality in patients with AMI, particularly among individuals with normal blood glucose levels upon admission to the ICU. Diabetes is a prevalent comorbidity among patients hospitalized with AMI, and its presence has been associated with increased in-hospital mortality [26]. Patients with diabetes typically exhibit a higher burden of atherosclerosis and a greater multivessel coronary artery disease [27]. Furthermore, individuals with AMI who experience diabetes often present with accelerated microvascular and macrovascular complications, which may contribute to their poorer outcomes [28]. In this study, we found that about 23% of AMI patients with diabetes experienced in-hospital mortality. However, the disparity in diabetes between survivors and non-survivors of AMI did not reach statistical significance (P>0.05). This may be related to the sample size.

This study explored the relationship between glycemic variability, glucose fluctuation trajectory and the risk of in-hospital mortality in patients with AMI, providing a clinical basis for blood glucose management and prognosis improvement among ICU patients with AMI. In addition, the eICU collaborative research database is a comprehensive collection of data from multiple centers, offering the representativeness of its samples. Of course, there are some limitations for this research. First, the generalizability of the study findings to other regions or countries may be limited due to the predominantly US-based study population. Second, there were inevitably some limitations in the number of blood glucose measurements and length of hospital stay, leading to potential selection bias. Third, as in other clinical studies, certain confounders, such as the Gensini score [29], were not accounted for in this study, potentially influencing in-hospital mortality in patients with AMI. Further prospective studies are still needed to validate our results.

## Conclusion

Glycemic variability was found to be correlated with in-hospital mortality, particularly among AMI patients who had normal blood glucose levels at admission. In addition, the presence of elevated blood glucose level at admission of AMI patients was observed as a significant risk factor for in-hospital mortality of AMI patients. Our study findings also suggest early intervention should be implemented to normalize high blood glucose levels at admission of AMI.

## Supporting information

**S1 Table. Sensitivity analysis of before and after interpolation.**
(DOCX)

**S2 Table. Screening of covariates.**
(DOCX)

## Author Contributions

**Conceptualization:** Junhua Chen, Nan Liang.

**Data curation:** Weifang Huang.

**Formal analysis:** Weifang Huang.

**Investigation:** Weifang Huang.

**Methodology:** Weifang Huang.

**Writing – original draft:** Junhua Chen, Nan Liang.

**Writing – review & editing:** Junhua Chen, Nan Liang.

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
