## [Decision Letter · Decision Letter 0]

22 Jan 2024

PONE-D-23-36914Blood glucose at admission, glycemic variability, blood glucose fluctuation and in-hospital mortality among patients with acute myocardial infarction: eICU Collaborative Research DatabasePLOS ONE

Dear Dr. Liang,

Thank you for submitting your manuscript to PLOS ONE. After careful consideration, we feel that it has merit but does not fully meet PLOS ONE’s publication criteria as it currently stands. Therefore, we invite you to submit a revised version of the manuscript that addresses the points raised during the review process.

We look forward to receiving your revised manuscript.

Kind regards,

Chikezie Hart Onwukwe

Academic Editor

PLOS ONE

Journal Requirements:

Reviewers' comments:

Reviewer's Responses to Questions

**Comments to the Author**

1. Is the manuscript technically sound, and do the data support the conclusions?

Reviewer #1: Partly

Reviewer #2: Yes

2. Has the statistical analysis been performed appropriately and rigorously? 

Reviewer #1: Yes

Reviewer #2: Yes

3. Have the authors made all data underlying the findings in their manuscript fully available?

Reviewer #1: Yes

Reviewer #2: Yes

4. Is the manuscript presented in an intelligible fashion and written in standard English?

Reviewer #1: Yes

Reviewer #2: Yes

5. Review Comments to the Author

Reviewer #1: Review of the manuscript: Blood glucose at admission, glycemic variability, blood glucose fluctuation and in-hospital mortality among patients with acute myocardial infarction: eICU Collaborative Research Database

This is a retrospective study that assessed the relationship between blood glucose levels at admission, (subsequent) glycemic variability, blood glucose fluctuation and the risk of in-hospital mortality in patients with acute myocardial infarction (AMI).

General comments:

1. The title and the whole content of the manuscript need some editing. For example, what is the technical difference between glycemic variability and blood glucose fluctuation? It appears in the manuscript as an artificial distinction and should be re-considered.

2. eICU was not defined or descibed clearly anywhere in the manuscript.

3. Ethical clearance issues: the authors indicate that requirement of ethical approval and informed, written consent were both waived for this particular work but it is not clear whether these were secured during the original data and sample collection and the construction of the database. The eICU database is deidentified – probably the authors are not aware of this – otherwise they need to state this clearly.

4. Please check a minor grammatical error on line 171.

Specific comments:

1. Novelty issue: Literature is awash with studies that have demonstrated that patients with AMI and hyperglycemia on admission have high rates of mortality. So, what is new in this particular work?

2. Scientific contribution:

a. the study doesn’t contribute towards resolution of the debate on whether or not acute hyperglycemia is causally related to the adverse outcomes after AMI.

b. the work, at best, corroborates the rich literature demonstrating the positive co-relationship between elevated blood glucose levels at admission with AMI and disease prognosis.

3. Unfounded claims: the authors state that “this is the first study on the relationship between blood glucose levels at admission, glycemic variability, blood glucose fluctuation and the risk of in-hospital mortality in patients with AMI” (from line 287 on). However, it is long established that there is a positive relationship between admission blood glucose levels and mortality after AMI. This is available in many of the articles they cited. Therefore, this needs to be corrected accordingly and the exact findings and novel contribution of their work, if any, should be clearly stated.

4. Effect of being diabetes: the data presented in Table 1 shows that diabetes is not significantly correlated with in-hospital survival, with only about 23% of patients not surviving. In fact, it is as if diabetic patients – expected to hyperglycemic unless properly managed – survive AMI better as compared to non-diabetics. I find it odd that this was not further discussed anywhere in the manuscript, given that glycemic variability was the main topic of the work.

In conclusion, the manuscript is better resubmitted after a major review, highlighting what novel finding was described here rather than the corroboration of a well-established finding. In addition, all of the other concerns indicated above need to be addressed.

Reviewer #2: Appreciate the great effort done by the authors and all the scientific process completed in this manuscript. It is an important daily medical issue that is simple to realize yet sometimes difficult to get under control thus evaluating this issue is always crucial for medical professionals. I don't have any additional comments.

6. PLOS authors have the option to publish the peer review history of their article (what does this mean?). If published, this will include your full peer review and any attached files.

Reviewer #1: No

Reviewer #2: No

---

## [Author Response · Author response to Decision Letter 0]

7 Feb 2024

Reviewers' comments:

Reviewer's Responses to Questions

Comments to the Author

1. Is the manuscript technically sound, and do the data support the conclusions?

Reviewer #1: Partly

Reviewer #2: Yes

Response: Thanks very much for your comments. Based on comments from all reviewers, we have revised the manuscript.

2. Has the statistical analysis been performed appropriately and rigorously?

Reviewer #1: Yes

Reviewer #2: Yes

Response: Thanks very much for your review.

3. Have the authors made all data underlying the findings in their manuscript fully available?

Reviewer #1: Yes

Reviewer #2: Yes

Response: Thanks very much for your comments.

4. Is the manuscript presented in an intelligible fashion and written in standard English?

Reviewer #1: Yes

Reviewer #2: Yes

Response: Thanks very much for your comments.

5. Review Comments to the Author

Reviewer #1: Review of the manuscript: Blood glucose at admission, glycemic variability, blood glucose fluctuation and in-hospital mortality among patients with acute myocardial infarction: eICU Collaborative Research Database

This is a retrospective study that assessed the relationship between blood glucose levels at admission, (subsequent) glycemic variability, blood glucose fluctuation and the risk of in-hospital mortality in patients with acute myocardial infarction (AMI).

General comments:

1. The title and the whole content of the manuscript need some editing. For example, what is the technical difference between glycemic variability and blood glucose fluctuation? It appears in the manuscript as an artificial distinction and should be re-considered.

Response: Thanks very much for your advice. We have modified the title as “Blood glucose fluctuation and in-hospital mortality among patients with acute myocardial infarction: eICU Collaborative Research Database”. The objective of this study was to analyze the correlation of glycemic variability and glucose fluctuation trajectory with the risk of in-hospital mortality in patients with AMI. Glycemic variability refers to the extent of fluctuations in blood glucose levels. While the glucose fluctuation trajectory in this study denotes the direction of changes in blood glucose levels, specifically, “Normal to Normal” means that both baseline and last measured blood glucose levels were “Normal”; “Intermediate to Intermediate” indicates that both baseline and last measured blood glucose levels were “Intermediate”; “High to High” means that both baseline and last measured blood glucose levels are “High”; “UP” indicates that the blood glucose level last measured was elevated compared to blood glucose level at admission; “High or Intermediate to Normal” indicates that the blood glucose level at admission was “High” or “Intermediate”, and the last measured blood glucose level was “Normal”; “High to Intermediate” means that the blood glucose at admission is “High” and the last measured blood glucose level is “Intermediate”. We have modified the content, and please see the revised manuscript. 

2. eICU was not defined or descibed clearly anywhere in the manuscript.

Response: Thanks very much for your advice. We have modified the content as following: Data for our analysis was sourced from the emergency intensive care unit (eICU) collaborative research database. The eICU collaborative research database is a large public database developed by the eICU Research Institute in collaboration with the Laboratory for Computational Physiology at Massachusetts Institute of Technology, which encompassing de-identified data from 200,859 intensive care unit (ICU) stays across 208 hospitals in the United States between 2014-2015.

3. Ethical clearance issues: the authors indicate that requirement of ethical approval and informed, written consent were both waived for this particular work but it is not clear whether these were secured during the original data and sample collection and the construction of the database. The eICU database is deidentified – probably the authors are not aware of this – otherwise they need to state this clearly.

Response: Thanks very much for your opinion. For this study, data for our analysis was sourced from the emergency intensive care unit (eICU) collaborative research database. The eICU collaborative research database is a large public database developed by the eICU Research Institute in collaboration with the Laboratory for Computational Physiology at Massachusetts Institute of Technology, which encompassing de-identified data from 200,859 intensive care unit (ICU) stays across 208 hospitals in the United States between 2014-2015. The utilization of this database has been granted approval by the Institutional Review Board at the Massachusetts Institute of Technology. All patient identity information is anonymized, thereby obviating the need for patient informed consent. Please see the revised manuscript.

4. Please check a minor grammatical error on line 171.

Response: We are grateful for the suggestion. We are sorry for our carelessness. We have modified the content as following: The non-survivors, in comparison to the survivors, exhibited advanced age and a higher proportion of congestive heart failure, atrial fibrillation, cardiogenic shock, sepsis. Additionally, they had higher levels of heart rate, respiratory rate, WBC, RDW, and creatinine.

Specific comments:

1. Novelty issue: Literature is awash with studies that have demonstrated that patients with AMI and hyperglycemia on admission have high rates of mortality. So, what is new in this particular work?

Response: We sincerely thank the reviewer for careful reading. Previous studies showed that admission hyperglycemia was associated with higher all-cause mortality in patients with AMI. However, intensive the insulin therapy is associated with an increased risk of unfavorable outcomes, potentially attributed to fluctuations in glucose levels and occurrences of hypoglycemia [1]. Furthermore, several research have indicated that focusing on blood glucose fluctuations may offer superior clinical management compared to solely monitoring [2, 3]. Glycemic variability, oscillations in blood glucose levels, refers to the measurement of fluctuations in glucose or other related parameters of glucose homeostasis over a specific time period [4]. An elevated glycemic variability can contribute to the activation of oxidative stress, impairment of endothelial function, and glycosylation of proteins [5-7]. No previously published study has examined the association between blood glucose fluctuations or variability and in-hospital mortality among patients diagnosed with AMI. Herein, the objective of this study was to analyze the correlation of glycemic variability and glucose fluctuation trajectory with the risk of in-hospital mortality in patients with AMI, exploring a reasonable range for short-term blood glucose control in patients with AMI and providing certain basis for treatment decision-making and prognosis improvement.

[1]. Bohé J, Abidi H, Brunot V, Klich A, Klouche K, Sedillot N, et al. Individualised versus conventional glucose control in critically-ill patients: the CONTROLING study-a randomized clinical trial. Intensive Care Med. 2021;47(11):1271-1283. Epub 2021/09/29. doi: 10.1007/s00134-021-06526-8. PMID: 34590159; PMCID: PMC8550173.

[2]. Sparks JR, Kishman EE, Sarzynski MA, Davis JM, Grandjean PW, Durstine JL, et al. Glycemic variability: Importance, relationship with physical activity, and the influence of exercise. Sports Med Health Sci. 2021;3(4):183-93. Epub 2022/07/06. doi: 10.1016/j.smhs.2021.09.004. PubMed PMID: 35783368; PubMed Central PMCID: PMCPMC9219280.

[3]. Psoma O, Makris M, Tselepis A, Tsimihodimos V. Short-term Glycemic Variability and Its Association With Macrovascular and Microvascular Complications in Patients With Diabetes. J Diabetes Sci Technol. 2022:19322968221146808. Epub 2022/12/29. doi: 10.1177/19322968221146808. PubMed PMID: 36576014.

[4]. Zhou Z, Sun B, Huang S, Zhu C, Bian M. Glycemic variability: adverse clinical outcomes and how to improve it? Cardiovasc Diabetol. 2020;19(1):102. Epub 2020/07/06. doi: 10.1186/s12933-020-01085-6. PubMed PMID: 32622354; PubMed Central PMCID: PMCPMC7335439.

[5]. Belli M, Bellia A, Sergi D, Barone L, Lauro D, Barillà F. Glucose variability: a new risk factor for cardiovascular disease. Acta Diabetol. 2023;60(10):1291-9. Epub 2023/06/21. doi: 10.1007/s00592-023-02097-w. PubMed PMID: 37341768; PubMed Central PMCID: PMCPMC10442283.

[6]. Klimontov VV, Saik OV, Korbut AI. Glucose Variability: How Does It Work? Int J Mol Sci. 2021;22(15). Epub 2021/08/08. doi: 10.3390/ijms22157783. PubMed PMID: 34360550; PubMed Central PMCID: PMCPMC8346105.

[7]. Valente T, Arbex AK. Glycemic Variability, Oxidative Stress, and Impact on Complications Related to Type 2 Diabetes Mellitus. Curr Diabetes Rev. 2021;17(7):e071620183816. Epub 2020/07/18. doi: 10.2174/1573399816666200716201550. PubMed PMID: 32674737.

2. Scientific contribution:

a. the study doesn’t contribute towards resolution of the debate on whether or not acute hyperglycemia is causally related to the adverse outcomes after AMI.

b. the work, at best, corroborates the rich literature demonstrating the positive co-relationship between elevated blood glucose levels at admission with AMI and disease prognosis.

Response: We are grateful for the suggestion. The objective of this study was to analyze the correlation of glycemic variability and glucose fluctuation trajectory with the risk of in-hospital mortality in patients with AMI. The findings indicated that glycemic variability was found to be correlated with in-hospital mortality, particularly in AMI patients with normal blood glucose levels at admission. In addition, the presence of elevated blood glucose level at admission of AMI patients was observed as a significant risk factor for in-hospital mortality. Our result also indicated that prompt management of elevated blood glucose levels upon admission and maintaining it within the normal range may be advantageous for the prognosis of patients with AMI. Please see the revised manuscript. 

3. Unfounded claims: the authors state that “this is the first study on the relationship between blood glucose levels at admission, glycemic variability, blood glucose fluctuation and the risk of in-hospital mortality in patients with AMI” (from line 287 on). However, it is long established that there is a positive relationship between admission blood glucose levels and mortality after AMI. This is available in many of the articles they cited. Therefore, this needs to be corrected accordingly and the exact findings and novel contribution of their work, if any, should be clearly stated.

Response: We sincerely thank the reviewer for careful reading. For this analysis, we aimed to explore the correlation of glycemic variability and glucose fluctuation trajectory with the risk of in-hospital mortality in patients with AMI. The findings indicated that highest quartile (CV>30) of CV were found to be positively correlated with in-hospital mortality, particularly in AMI patients with normal blood glucose levels at admission. In addition, the presence of elevated blood glucose level at admission of AMI patients was observed as a significant risk factor for in-hospital mortality. Our result also indicated that prompt management of elevated blood glucose levels upon admission and maintaining it within the normal range may be advantageous for the prognosis of patients with AMI.

4. Effect of being diabetes: the data presented in Table 1 shows that diabetes is not significantly correlated with in-hospital survival, with only about 23% of patients not surviving. In fact, it is as if diabetic patients – expected to hyperglycemic unless properly managed – survive AMI better as compared to non-diabetics. I find it odd that this was not further discussed anywhere in the manuscript, given that glycemic variability was the main topic of the work.

Response: We are grateful for the suggestion. We have added the content in the Discussion section: Diabetes is a prevalent comorbidity among patients hospitalized with AMI, and its presence has been associated with increased in-hospital mortality [1]. Patients with diabetes typically exhibit a higher burden of atherosclerosis and a greater multivessel coronary artery disease [2]. Furthermore, individuals with AMI who experience diabetes often present with accelerated microvascular and macrovascular complications, which may contribute to their poorer outcomes [3]. In this study, we found that about 23% of AMI patients with diabetes experienced in-hospital mortality. However, the disparity in diabetes between survivors and non-survivors of AMI did not reach statistical significance (P>0.05). This may be related to the sample size. Further prospective studies are still needed to validate our results.

[1] Baviera M, Genovese S, Colacioppo P, Cosentino N, Foresta A, Tettamanti M, et al. Diabetes mellitus duration and mortality in patients hospitalized with acute myocardial infarction. Cardiovasc Diabetol. 2022;21(1):223. Epub 2022/10/31. doi: 10.1186/s12933-022-01655-w. PMID: 36309742; PMCID: PMC9618227.

[2] Kim CS, Choi JS, Park JW, Bae EH, Ma SK, Jeong MH, et al. Concomitant renal insufficiency and diabetes mellitus as prognostic factors for acute myocardial infarction. Cardiovasc Diabetol. 2011;10:95. Epub 2011/11/01. doi: 10.1186/1475-2840-10-95. PubMed PMID: 22035298; PubMed Central PMCID: PMCPMC3225317.

[3] Alabas OA, Hall M, Dondo TB, Rutherford MJ, Timmis AD, Batin PD, et al. Long-term excess mortality associated with diabetes following acute myocardial infarction: a population-based cohort study. J Epidemiol Community Health. 2017;71(1):25-32. Epub 2016/06/17. doi: 10.1136/jech-2016-207402. PubMed PMID: 27307468.

In conclusion, the manuscript is better resubmitted after a major review, highlighting what novel finding was described here rather than the corroboration of a well-established finding. In addition, all of the other concerns indicated above need to be addressed.

Response: We are grateful for the suggestion. The findings indicated that highest quartile (CV>30) of CV were found to be positively correlated with in-hospital mortality, particularly in AMI patients with normal blood glucose levels at admission. In addition, the presence of elevated blood glucose level at admission of AMI patients was observed as a significant risk factor for in-hospital mortality. Our result also indicated that prompt management of elevated blood glucose levels upon admission and maintaining it within the normal range may be advantageous for the progn

---

## [Decision Letter · Decision Letter 1]

27 Feb 2024

Blood glucose fluctuation and in-hospital mortality among patients with acute myocardial infarction: eICU Collaborative Research Database

PONE-D-23-36914R1

Dear Dr. Nan Liang,

We’re pleased to inform you that your manuscript has been judged scientifically suitable for publication and will be formally accepted for publication once it meets all outstanding technical requirements.

Kind regards,

Chikezie Hart Onwukwe

Academic Editor

PLOS ONE

Additional Editor Comments (optional):

Reviewers' comments:

Reviewer's Responses to Questions

**Comments to the Author**

1. If the authors have adequately addressed your comments raised in a previous round of review and you feel that this manuscript is now acceptable for publication, you may indicate that here to bypass the “Comments to the Author” section, enter your conflict of interest statement in the “Confidential to Editor” section, and submit your "Accept" recommendation.

Reviewer #1: All comments have been addressed

2. Is the manuscript technically sound, and do the data support the conclusions?

Reviewer #1: Yes

3. Has the statistical analysis been performed appropriately and rigorously? 

Reviewer #1: I Don't Know

4. Have the authors made all data underlying the findings in their manuscript fully available?

Reviewer #1: Yes

5. Is the manuscript presented in an intelligible fashion and written in standard English?

Reviewer #1: Yes

6. Review Comments to the Author

Reviewer #1: I think the authors have sufficiently addressed my concerns. Therefore, I do not object to the publication of their work if the editors wish to do so.

7. PLOS authors have the option to publish the peer review history of their article (what does this mean?). If published, this will include your full peer review and any attached files.

Reviewer #1: No
